# EXPLORING UNFAIRNESS IN INTEGRATED GRADIENTS BASED ATTRIBUTION METHODS

## ABSTRACT

Numerous methods have attempted to explain and interpret predictions made by machine learning models in terms of their inputs. Known as "attribution methods" they notably include the Integrated Gradients method and its variants. These are based upon the theory of Shapley Values, a rigorous method of fair allocation according to mathematical axioms. Integrated Gradients has axioms derived from this heritage with the implication of a similar rigorous, intuitive notion of fairness. We explore the difference between Integrated Gradients and more direct expressions of Shapley Values in deep learning and find Integrated Gradients' guarantees of fairness weaker; in certain conditions it can give wholly unrepresentative results. Integrated Gradients requires a choice of "baseline", a hyperparameter that represents the 'zero attribution' case. Research has shown that baseline choice critically affects attribution quality, and increasingly effective baselines have been developed. Using purpose-designed scenarios we identify sources of inaccuracy both from specific baselines and inherent to the method itself, sensitive to input distribution and loss landscape. Failure modes are identified for baselines including Zero, Mean, Additive Gaussian Noise, and the state of the art Expected Gradients. We develop a new method, Integrated Certainty Gradients, that we show avoids the failures in these challenging scenarios. By augmenting the input space with "certainty" information, and training with random degradation of input features, the model learns to predict with varying amounts of incomplete information, supporting a zero-information case which becomes a natural baseline. We identify the axiomatic origin of unfairness in Integrated Gradients, which has been overlooked in past research.

## 1 INTRODUCTION

Attribution, the identification of the input features most salient to a model prediction, is an increasingly important requirement for neural networks. It is a core part of model interpretability, which is valuable as a research tool and design aid, but also increasingly as an output requirement in its own right for uses as diverse as medical imaging (Singh et al., 2020) to loan applications (Bhatt et al., 2020). A "right to explanation" of machine decisions is even provided in the European Union's General Data Protection Regulation (Goodman & Flaxman, 2017).

Gradients of predictions with respect to model inputs can be calculated using backpropagation and have been used for feature attribution ("Vanilla Gradient") (Erhan et al., 2009; Simonyan et al., 2014; Yosinski et al., 2015). The gradients indicate which features are most sensitive to a perturbation, causing the largest change in prediction outcome. Empirically, this method often does highlight areas relevant to prediction, but it may also miss important areas due to the nonlinearity of the model and saturation of the gradients (Sundararajan et al., 2016).

The method of *Integrated Gradients* (Sundararajan et al., 2016; 2017) overcomes these issues by generating attributions based on the theory of Shapley values (Sundararajan et al., 2017). The method integrates the gradients as the input varies linearly between a *baseline* and the final input of interest:

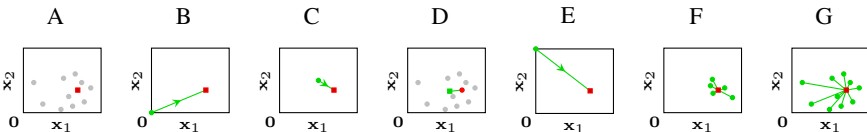

Figure 1: Examples of (generalized) baselines. Grey: input distribution; Red: attributed input; Green: baselines and integration paths. A: Distribution and target input. Baselines: B: Zero; C: Middle; D: Distribution mean; E: Maximum distance; F: Additive Gaussian noise; G: Expected Gradients. Failure modes: B-D are single static baselines so exhibit baseline blindness. Maximum Distance gives poor attributions due to extreme O.O.D. baselines. AGN fails the Local Minimum scenario (section 5.2). EG fails the Burnt Snacks scenario (section 5.1).

$$\text{Integrated Gradients}(F, \boldsymbol{x}', \boldsymbol{x}, i) := \underbrace{(\boldsymbol{x}_i - \boldsymbol{x}_i')}_{\text{distance}} \underbrace{\int_{\alpha=0}^{1} \frac{\partial F(\boldsymbol{x})}{\partial \boldsymbol{x}_i}\bigg|_{\boldsymbol{x} = \boldsymbol{x}' + \alpha(\boldsymbol{x} - \boldsymbol{x}')} d\alpha}_{\text{mean gradient}} \quad (1)$$

where $F(\cdot)$ denotes the prediction of the model, $\boldsymbol{x}$ is the input vector of interest, $\boldsymbol{x}'$ is the baseline input vector, and $i$ indexes to the feature of interest. A numerical approximation is normally used to calculate the integral in practice. The baseline represents a *missing* or *neutral* input, a concept required by the Shapley values theory, but shared with several other attribution methods (Covert et al., 2020).

In the remainder of this paper, we first give an overview of baselines discussed in the literature (Section 2). Then we discuss the theoretical cause of a failure mode of IG (*attribution transfer*) using a zero baseline example (Section 3). In Section 4 we introduce a new method, *Integrated Certainty Gradients*. Finally we show in experiments that other baselines including the state of the art Expected Gradients do not prevent attribution transfer, while in the tested scenarios Integrated Certainty Gradients does, and discuss additional failure cases of specific baselines (Section 5).

**Note on notation** It will be convenient to imply an equivalence between attribution methods which take a baseline, $\varphi(F, \boldsymbol{x}', \boldsymbol{x}, i)$, and those which do not $\varphi(F, \boldsymbol{x}, i)$. A baseline free method can be adapted to take a baseline by $\varphi_B(F, \boldsymbol{x}', \boldsymbol{x}, i) = \varphi(F, \boldsymbol{x}, i) - \varphi(F, \boldsymbol{x}', i)$.

## 2 CHOICE OF BASELINE

Baseline choice has a major effect on attribution outcome. Effective choice of baselines in practice is a topic of ongoing research. A recent review of four state-of-the-art models for tabular data did not find a best performing choice (Haug et al., 2021). As this paper noted, the concept of missingness within an arbitrary space is domain-specific. Therefore applying attribution techniques to a new domain requires this concept to be determined. Another review came to similar conclusions for image classification attribution (Sturmfels et al., 2020).

Baselines can be generalized from single inputs to distributions by taking the expected value of Integrated Gradients using each input in the distribution. In practice $X'$ is often from an empirical (discrete uniform) distribution so the expectation becomes simply the mean over $X'$. A summary of common baselines used in Integrated Gradients and other attribution methods follows. For a more comprehensive review, one can refer to Haug et al. (2021); Sturmfels et al. (2020). Visualizations of integration paths for various baselines are shown in Figure 1.

**Constant value** baselines include the uniform black/zero baseline proposed with the original method (Sundararajan et al., 2016) and choices that attempt to be unbiased with respect to the input distribution, including the component-wise mean of the input distribution, the middle of the input range, and uniform noise. If the input matches the baseline, the $\boldsymbol{x}_i - \boldsymbol{x}_i'$ term in Equation 1 disappears, any such areas will not be attributed, even when they are essential to the model's prediction ("baseline blindness") (Sturmfels et al., 2020). When the baseline is constant this undesirable behavior cannot be avoided. This behavior can be seen clearly in attributions of MNIST images with a zero baseline, for example in Adebayo et al. (2020).

**Distribution baselines** avoid baseline blindness by combining multiple attributions. Uniform noise has been used in this way, although concerns have been identified about the risk of interaction between the baseline and high frequency features (Sundararajan et al., 2017; Sturmfels et al., 2020). The recently developed Expected Gradients method (Erion et al., 2020) uses the input (typically training) distribution as the baseline. It computes the attribution for a given input example efficiently by sampling baselines randomly from the training distribution, and interpolation weights from the the [0, 1] range (Monte Carlo integration), rather than using the many equally spaced interpolations of the more common Riemann Sum method.

**Dynamic baselines** change based on the attribution input. The maximum distance baseline uses the farthest point in the input space from the input vector (Sturmfels et al., 2020). This avoids baseline blindness but abandons the concept of neutrality. Sturmfels et al. (2020) cite Fong & Vedaldi (2017) as the inspiration for blurring as a possible method of feature ablation to produce a baseline. Gaussian noise may be added to the input vector during attribution and when averaged over several iterations, the resulting attributions were found to be more detailed and less noisy (Smilkov et al., 2017). Adding noise to input vectors during training was also found to give a similar and cumulative effect. The combination of input vector with noise can be used by itself as a dynamic distribution baseline (Sturmfels et al., 2020) or as an augmentation technique in combination with another baseline. The justification of this method for use with attribution appears to be mainly empirical, and a question regarding the theoretical justification has been raised (Sturmfels et al., 2020).

## 3 ATTRIBUTION FAIRNESS

Assessment of attribution methods is not currently straightforward because there is not yet a unified general procedure to do so. Rather, there are a number of tests for various indicative characteristics (Hooker et al., 2019; Adebayo et al., 2018; Yang & Kim, 2019). We focus our attention on fairness: whether the relative attributions of input components are reasonable. This is the central concern of Shapley Values, a celebrated method from game theory for allocating value, with a strongly justified mathematical foundation (Shapley, 1952), and the background to the Integrated Gradients methods (Sundararajan et al., 2017).

Assessing fairness is not trivial. As a reference we consider two other attribution methods defined using the original Shapley Values procedure: BShap (Sundararajan & Najmi, 2020) and SHAP (Lundberg & Lee, 2017). Shapley Values is a procedure that, for a given set $S$, requires a value for every possible subset of elements: $v : \mathcal{P}(S) \to \mathbb{R}$, where $\mathcal{P}$ is the power set of $S$, and gives back values for the elements themselves (the "Shapley values"): $\phi_{\text{sv}}(v, e)$. The Shapley values sum to the value of the complete set: $v(S) = \sum_{e \in S} \phi_{\text{sv}}(v, e)$, and certain axioms, widely considered as naturally related to fairness, are maintained. As such it gives a way of sharing the value of the complete set fairly among the elements, based on the values that are achieved by each subset.

Model inputs cannot typically represent missing components, so to allocate attribution to the components of an input vector using Shapley Values, presence/absence of a component must be replaced by another binary property. In the case of BShap, 'present' and 'absent' components are represented with values taken from the equivalent components of the input vector $\boldsymbol{x}$ or a baseline $\boldsymbol{x}'$ respectively. The value function $v$ is simply the model's prediction $F$. SHAP represents subsets using random vectors, and the value function $v$ by the expectation of the model's prediction $\mathbb{E} \circ F$. Each vector has a distribution $f_{(\boldsymbol{X} | \boldsymbol{X}_i = \boldsymbol{x}_i \forall i)}$ where $\boldsymbol{X}$ is the distribution of possible inputs, $\boldsymbol{x}$ is the input being attributed, and $i$ are the 'present' components of the vector. Further information about Shapley Values and these methods is given is Section A.1 of the supporting material. To summarize the relationships:

| | | | | | |
|---|---|---|---|---|---|
| $\phi_{\text{sv}}(v, e)$ | $v$ | sets | elements $e$ | $S$ | $\{\}$ |
| $\varphi_{\text{BShap}}(F, \boldsymbol{x}', \boldsymbol{x}, i)$ | $F$ | vectors | components $i$ | $\boldsymbol{x}$ | $\boldsymbol{x}'$ |
| $\varphi_{\text{SHAP}}(F, \boldsymbol{x}, i)$ | $\mathbb{E} \circ F$ | random vectors | components $i$ | $\mathbf{a} = \boldsymbol{x}$ | $\mathbf{a} \sim \mathbf{X}$ |

### 3.1 ATTRIBUTION TRANSFER

Integrated Gradients integrates partial derivatives along a straight path between a baseline and the input vector. This can be roughly imagined as accumulating the contours $\partial f(\boldsymbol{x})$ of the input-output landscape that fall across the integration path. To explore fairness in IG we consider the scenario shown in Figure 2. The scenario is to classify whether either of two components $\boldsymbol{x}_1, \boldsymbol{x}_2$ fall within

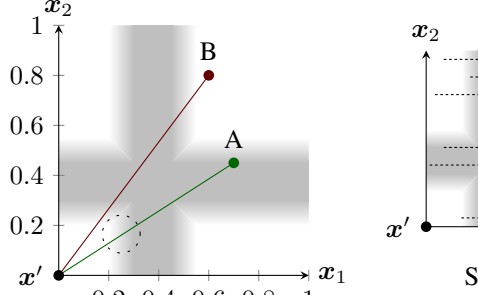 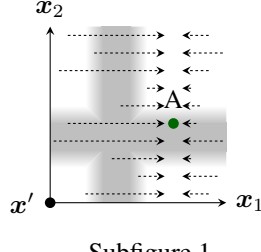 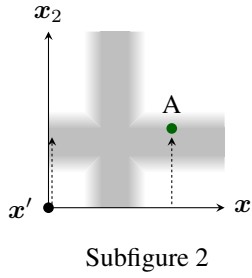

Figure 2: Representation of a model that indicates whether either component of the input $\boldsymbol{x} \in [0,1]^2$ falls within the approximate range 0.25 - 0.5. The background color indicates output value, from white (false, 0) to grey (true, 1). Two IG integration paths exhibiting attribution transfer are shown. Both paths start from baseline $\boldsymbol{x}' = (0,0)$. Path A indicates an input $\boldsymbol{x}$ with $\boldsymbol{x_2}$ in the target range and $\boldsymbol{x_1}$ outside. However since the integration path crosses only a 'horizontal' gradient (indicated in the dashed circle) IG will incorrectly attribute the result (class 1) to $\boldsymbol{x_1}$. There is no possible change in $\boldsymbol{x_1}$ alone that can account for the increased output of A (Subfigure 1). Conversely $\boldsymbol{x_2}$ alone can account for the full increase by changing at $\boldsymbol{x}'$, $\boldsymbol{x}$, or many other places (Subfigure 2). Hence the IG result could arguably be considered 'maximally misleading'. Path B indicates an input $\boldsymbol{x}$ with both components above the target range. As the path crosses perpendicular gradients, both components are attributed with opposite signs.

Table 1: Attribution results associated with Figure 2. IG and BShap have baseline at $(0, 0)$. SHAP values have a different sum because SHAP effectively uses the expectation across the whole input space as the baseline. The input distribution is assumed uniform for SHAP. Calculations are given in section A.1.3 of the supporting material.

|   | BShap | SHAP | IG |
|---|---|---|---|
| A | (0, 1) | (-0.11, 0.66) | (1, 0) |
| B | (0, 0) | (-0.22, -0.22) | (-1, 1) |

the range 0.25 - 0.5, and the shown model gives a continuous approximation of this (transitioning linearly from 0.05 either side of the boundary). The figure shows IG attribution paths for two inputs A and B. Attribution results for these inputs are shown in Table 1.

The results for path A could be considered 'maximally misleading' as discussed in Figure 2. The other methods in Table 1 are consistent with this. To understand why IG gives this result, we compare the scenario with a case where there is a ground truth attribution. For additive models $f(\boldsymbol{x}) = \sum_i f_i(\boldsymbol{x_i})$ each component's contribution can be considered independently: $\varphi(f, \boldsymbol{x}', \boldsymbol{x}, i) = f_i(\boldsymbol{x_i}) - f_i(\boldsymbol{x_i}')$. Linear models are a subset. In this regime BShap and IG give matching results, as does SHAP if the baselines $\boldsymbol{x}'$ were chosen such that $f_i(\boldsymbol{x_i}') = \mathbb{E}(f_i)$ and the input distribution is uniform. The model in Figure 2 deviates from additivity where the target regions of each component overlap. By passing through this area the integration path can enter the target region of $\boldsymbol{x_2}$ without crossing any 'horizontal' contours ($\frac{\partial f(\boldsymbol{x})}{\partial \boldsymbol{x_2}} > 0$), yielding an unintuitive result. We name this phenomenon "attribution transfer": the change in attribution caused by the integration path crossing a region not representative of the wider landscape. This is a precise concept when the model is mostly additive. For general models, as it becomes harder to say what is "representative" this becomes less precisely defined, although it is still useful for discussing the effect of specific perturbations in the landscape, for example.

Path B attributes an input vector with both components above the target range. IG explains the zero result as the balance of two competing contributions, unlike BShap and Shap which consider the components to contribute equally. Due to sharp features of the input-output landscape near the integration path, the IG attribution will undergo an abrupt reversal as the input crosses the $\boldsymbol{x_1} = \boldsymbol{x_2}$ boundary, even though neither the input nor baseline are near a decision boundary, nor in the target range. IG attribution results can be arbitrarily affected by small sharp regions that fall across the path.

Is it reasonable to be concerned these effects may occur in practice? We feel it is given that gradients are known to fluctuate sharply (Smilkov et al., 2017), in addition to the results we present.

## 3.2 AXIOMATIC FOUNDATIONS

Like the other methods, IG is based on Shapley theory and has rigorous axioms. It might then be surprising that IG permits 'maximally misleading' results. IG is based on a method in the Shapley Values lineage known as Aumann-Shapley (Aumann & Shapley, 1974), and has similar but not identical axioms. The names of the axioms of IG and BShap include those of the original Shapley Values: *Efficiency*, *Linearity*, *Symmetry*, and *Dummy* (Sundararajan & Najmi, 2020). However, IG takes a more liberal generalization of the Dummy axiom. For clarity we will give separate names to each variant:

**Definition 1** (Weak Dummy). An attribution method $\varphi$ satisfies Weak Dummy iff

$$f(\boldsymbol{z}) = f(\boldsymbol{z} + \alpha \mathbf{1}_i) \, \forall \boldsymbol{z}, \alpha \Rightarrow \varphi(f, \boldsymbol{x}, i) = 0$$

This simply means that if the model does not use an input feature, it has zero attribution.

**Definition 2** (Strong Dummy (BShap)). Let $\boldsymbol{a}$ be a vector with each component $\boldsymbol{a}_k$ drawn from the corresponding component of either of two given vectors: $\boldsymbol{x}_k$ or $\boldsymbol{x}'_k$, with $\boldsymbol{a}_i = \boldsymbol{x}_i$ for a given $i$. Let $\boldsymbol{b} = \boldsymbol{a}$ except that $\boldsymbol{b}_i = \boldsymbol{x}'_i$. Then an attribution method $\varphi$ satisfies Strong Dummy iff $f(\boldsymbol{a}) - f(\boldsymbol{b}) = 0 \, \forall \boldsymbol{a} \Rightarrow \varphi(f, \boldsymbol{x}', \boldsymbol{x}, i) = 0$

Strong Dummy is identical to Weak Dummy except that only component values from the target and baseline vector are considered, not the whole range. The weaker conditions make the axiom stronger. Both IG and BShap are formulated to satisfy Weak Dummy, but BShap additionally possesses Strong Dummy. This precludes it from exhibiting attribution transfer and from producing unintuitive attributions of the kind given by IG in Table 1. The situation indicated in Figure 2 Subfigure 1 is prohibited; any positively attributed component must be responsible for an increase in output value relative to another location in the input space (proof in Section A.2 of the supporting material). Attribution transfer is not possible because it requires sensitivity to the region between the baseline and target value. Conversely, Weak Dummy guarantees that a component does not participate in attribution transfer only if it has no influence on the model at all. We feel a clear distinction between the axioms in the (original) discrete case and the continuous case may be helpful to avoid any misconception about the strength of the guarantees provided, which are considered extremely robust for Shapley Values in their original context.

## 4 INTEGRATED CERTAINTY GRADIENTS

We have shown that certain features of the input-output landscape can lead to unintuitive IG attributions. To contrast this behavior in our experiments we introduce a new IG based method, "Integrated Certainty Gradients", which does not traverse the original input space, but rather the space of conditional probabilities, in a manner related to SHAP. Sundararajan & Najmi (2020) note that SHAP ("CES" in their work) is not immediately applicable on finite continuous-valued distributions (such as a dataset) due to their feature sparsity. Our method resolves this by having the model learn a generalization of the dataset. As a Shapley Values method, direct calculation of SHAP is intractable for high dimensional data, and our method is an Aumann-Shapley alternative to SHAP in the same way IG is for BShap.

Our method adds additional information, the *certainty*, $\boldsymbol{c}$, to the model input, so the model bases its prediction on the input and the *certainty* together: $(\boldsymbol{x}, \boldsymbol{c})$. The dimension of $\boldsymbol{c}$ is the same as the number of input features in $\boldsymbol{x}$, and it takes values between $[0, 1]$ that indicate the amount of information the corresponding feature provides. The per-pixel, per-input certainty $c$ is drawn from a random distribution throughout the training of a model, and the model must be trained to associate low certainty values with a concept of *missingness* suitable for the Shapley theory. This is done using a training process that applies a *damage function* $D$ to the input images, so the model receives *damaged inputs*: $(D(\boldsymbol{x}, \boldsymbol{c}), \boldsymbol{c})$. A diagram showing the *damage* process is given in Figure 3. First, *damage data* is required, of the same dimension as the undamaged input $\boldsymbol{x}$. This can take many forms, but for now we will use random noise where each component is i.i.d. uniformly in $[0, 1]$. Then a *damage mask* is generated by treating each component $\boldsymbol{c}_i$ as the probability of keeping the

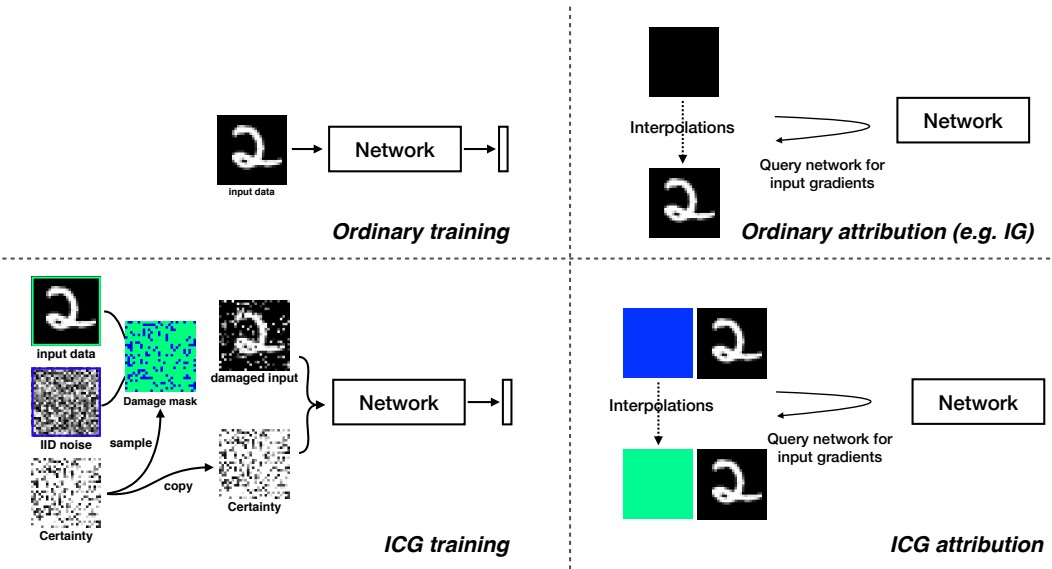

Figure 3: Visualisation of ordinary vs. Integrated Certainty Gradients (ICG) training and attribution. The top row shows ordinary training and attribution using Integrated Gradients which interpolates between a baseline (here: black) and the input example. The bottom row shows our proposed ICG approach. Blue and green correspond to damaged and undamaged components (left side), and certainty 0 and 1 (right side). During training, we sample a certainty vector, then sample damage using the probabilities from the certainty vector. The model takes this certainty vector as input and thus learns how to account for probabilistically reliable input features. ICG attribution is then simply the Integrated Gradient from no information to full information about an input example.

corresponding feature $x_i$. If $x_i$ is not kept it is replaced the corresponding value from the *damage data*.

A method is needed to generate $c$ during training. Our method works reasonably well if each component $c_i$ (and thus the keep probability of each input feature) is i.i.d. uniformly in $[0, 1]$. Regular (undamaged) inputs to the model correspond to using $c_i = 1 \forall i$. The features $D(x, c)_i$ of an input prepared in this way have varying chances, equal to their certainties $c_i$, to provide information about the original input $x$, and therefore the class $\hat{y}$. During training the model learns to ignore features unrelated to $\hat{y}$, discounting features with lower certainty, and develops the required uncertainty semantics.

If the model is trained well, it will ignore features with zero certainty, because these give no information about the original input:

$$\forall\, i : \left. \frac{\partial F(x, c)}{\partial x_i} \right|_{c_i = 0} \approx 0 \tag{2}$$

In this case, $c = 0$ provides a natural baseline to represent missing data, as the model will ignore the input values entirely.

We complete the Integrated Certainty Gradients attribution method by combining this baseline with IG. The integral of Equation 1 is taken from $c = 0$ to $c = 1$, holding $x$ constant at the input we wish to attribute. Note that $x$ is *not* damaged while calculating the attribution. It is convenient to note when implementing that the distance term from Equation 1 disappears because $c_i - c_i' = 1$.

Deep networks are capable of detecting and ignoring noise, so the *damage data* described above may not guarantee learning a correct interpretation of certainties. To avoid this, samples from the input dataset can be used as *damage data*, ensuring certainties must be used to distinguish original and damaged features. During ICG the model will encounter interpolations with total mean *certainty* from 0 to 1. Most inputs generated with *certainty* sampled I.I.D. from U(0, 1) will have mean *certainty*

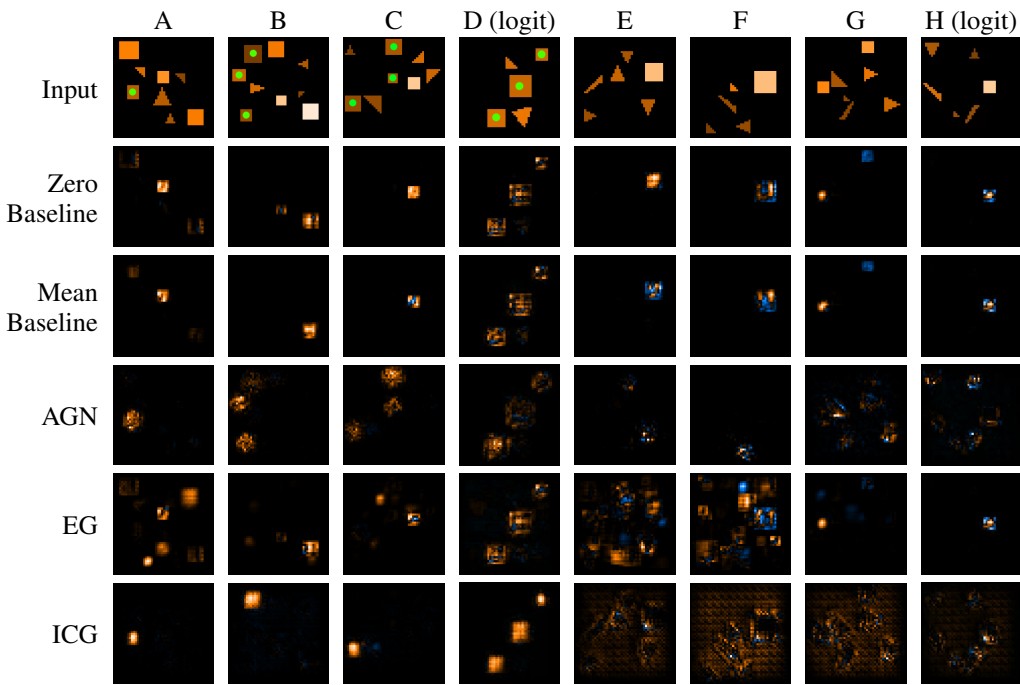

Figure 4: Attribution results for the Burnt Snacks scenario. Each column shows an input image (top row) with associated attributions (other rows). Column A-D inputs are randomly chosen from class 1 (dark square present). Column E-G inputs are randomly chosen from class 0 (no dark square). Probabilities are attributed except columns D and H where logits are attributed. The target objects (dark squares) are marked with a green dot (not present in actual input vector). Expected Gradients is evaluated with 500 interpolations. Additive Gaussian Noise is evaluated with 50 baselines with 100 interpolations each. The other methods are evaluated with 100 interpolations.

close to 0.5. To ensure the model is trained on a wider range of *certainty* inputs we developed a parameterized distribution based on the Continuous Bernoulli (Loaiza-Ganem & Cunningham, 2019) where the mean can be varied betwen 0 and 1. Full information about the training methods used is given in Section A.3.2 of the supporting material.

# 5 EXPERIMENTS

## 5.1 BURNT SNACKS SCENARIO

We use a purpose designed scenario to demonstrate attribution transfer of methods including Expected Gradients. The scenario is notionally similar to an Optical Sorting task in which overcooked food items are identified (Pedreschi et al., 2016). The model must determine whether 'overcooked' (dark) squares (25% - 50% lightness) are present (class 1), or not (class 0), amidst light squares and other shapes on a dark background. The input-output landscape for an input with two squares with lightnesses $x_1$, $x_2$ is similar to Figure 2. Two characteristics are designed to mislead Expected Gradients. First, the average pixel value is very dark. This will tend to darken the features of an attributed image when it is interpolated. Second, the light squares are lighter versions of the target objects (dark squares), but are not relevant to the prediction outcome. This combination will cause light squares to often present as dark squares in interpolations, despite having no correlation with them, confusing Expected Gradients. It is easy to justify that light squares should not be attributed, because their presence does not influence the class (relative to black background). A selection of results on this scenario are shown in Figure 4. We trained a certainty aware model to 98.7% accuracy (for undamaged inputs). We attribute probabilities, in line with a prominent guide (Google, 2021), and also logits.

**Zero and mean baseline**  Although a zero baseline seems appropriate, both zero and mean baselines markedly show attribution transfer. Example G for the zero baseline case is equivalent to path B of Figure 2. There are two light squares, one with negative and one positive attribution. By checking the interpolation gradients, we saw the positive attributed square enter the target region (at 25% lightness) first, picking up attribution. The second square followed it but gathered no attribution due to saturation of the output. On leaving (at 50% lightness) the situation was reversed. Attribution transfer also occurs in class 1 scenarios where a light square is present, for example Zero Baseline example C. This situation is equivalent to path A of Figure 2. As a result, the process concludes with only the light square, irrelevant to the prediction, attributed.

**Expected Gradients**  Results for Expected Gradients are shown in the 5th column of Figure 4. In no case does EG identify the dark squares when they are present when using probabilities (class 1, images A-C). Instead, it transfers attribution to the light squares, most strongly the lightest. In case A it appears no square is light enough to attract attribution, and artifacts from the random sampling process are shown.

Since class 0 is an absence class the expected outcome is not as clear. Images E and F show diffuse artifacts from the interpolation process and arguably this is a fair result. However image G strongly attributes a specific square and this seems more clearly misleading, as the square has no significance to the classification outcome.

**Integrated Certainty Gradients**  ICG does not suffer from the previously described problems. Because the integration is across the uncertainty space, the relative location of features in the input space and the input distribution are not important. For class 0, ICG attributions are very diffuse and largely uniform, which could be considered an appropriate outcome, arguably the most intuitive of any method. For class 1, it always attributes a dark square (correct target feature). The attributions are strong and precisely located. However, only one square is attributed even when multiple are present. By inspecting the interpolations and using variations of the image with each square removed, we found that the model's confidence in each square transitions abruptly during the certainty increase. Because of this, the first square to gain the model's confidence causes the prediction to saturate, preventing it accruing to other squares. This is attribution transfer in the certainty space. This shows that though irrelevant objects are no longer brought into in the attribution, there can still be transfer between independent relevant objects. We leave further consideration of this aspect to future research.

**Additive Gaussian Noise**  Additive Gaussian Noise correctly attributes the location of the dark squares in class 1 images. The attributions are noisy and not as precisely localised to the squares as those of ICG, however, all squares are attributed. In class 0 attribution focuses around the triangles. This is unsurprising because the addition of noise could easily make them resemble squares.

**Logits**  For most methods logits show intermixed opposite attributions on both light and dark squares. Dark squares have overall more positive attribution, but nonetheless this presentation is imperfect and potentially confusing. ICG performs excellently in this scenario, no longer missing squares, attributing all dark squares clearly.

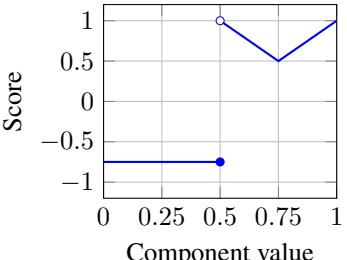

## 5.2  Local Minimum scenario

Additive Gaussian Noise uses baselines drawn mainly from a region local to the input vector. We introduce a scenario designed to challenge local baselines. Input vectors are classified according to a total score $S(\boldsymbol{x})$, which is the sum of component-wise scores $f(\boldsymbol{x}_i)$. Inputs are in class 1 if $S(\boldsymbol{x}) > 0$, class 0 otherwise. Note that although $S$ is an additive function, the classifier as a whole is not. The component-wise score function $f$ is shown in Figure 5. $f$ has a positive local minimum. This local minimum makes a positive contribution to class 1 even though it is less contributing than nearby values. Attribution methods based on local baselines would be expected to give the local minimum an incorrect negative attribution. We trained a certainty-aware network on this classification scenario to 93%

Figure 5: Component-wise score function $f$ for the Local Minimum scenario. The total score is the sum of the scores of the covariates $S(\boldsymbol{x}) = \sum_i f(\boldsymbol{x}_i)$. Component values range from 0 to 1. The score is $-0.75$ for component values 0.5 and below, and between 0.5 and 1 for values above 0.5.

Table 2: Comparison of attribution methods on the Local Minimum scenario. Each row corresponds to a component of the input vector. The first column shows the input vector. The second column shows the nominal "scores" of the components contributing to the true class, normalised to maximum value 1, for comparison with attribution results. The remaining columns show attribution results, normalised to maximum value 1, from BShap, zero baseline Integrated Gradients, mean baseline Integrated Gradients, Expected Gradients, Additive Gaussian Noise and Integrated Certainty Gradients. Attributions are shown for class 1, not the predicted class (0), to aid comparison with the score values.

| Input vector | Scaled score | BShap | Zero IG | Mean IG | EG | AGN | ICG |
|---|---|---|---|---|---|---|---|
| 0.1 | -0.94 | -0.75 | 0.001 | -0.01 | -0.30 | 0.004 | -0.75 |
| 0.2 | -0.94 | -0.74 | 0.002 | -0.001 | -0.73 | -0.004 | -0.75 |
| 0.3 | -0.94 | -0.74 | 0.003 | -0.37 | -0.66 | 0.004 | -0.74 |
| 0.6 | 1 | 1 | 1 | 1 | 0.82 | 1 | 1 |
| 0.7 | 0.75 | 0.50 | -0.24 | 0.89 | 0.70 | -0.42 | 0.48 |
| 0.9 | 1 | 0.76 | 0.23 | -0.07 | 1 | -0.02 | 0.74 |

accuracy. Attribution results for this network using an example input vector from class 0 are shown in Table 2. Component values of 0.50546 were used for the BShap baseline, as this was determined to be close to the model's prediction boundary.

BShap, ICG and EG performed well in the scenario, with ICG and BShap in excellent agreement. Only IG and BShap accurately represent the uniform negative region. All other methods performed badly. As expected, AGN incorrectly gives the local minimum a negative attribution. Zero IG shows strong signs of attribution transfer. Zero IG and AGN both exhibit blindness in the negative score region due to sharing this negligible gradient region with the baselines. The results for mean IG vary widely. The shown result is a randomly chosen example. The distribution mean lies on the discontinuity of the score function (0.5) so small variations around it make major differences to the result.

## 6 CONCLUSION

We have described a situation in which Integrated Gradients produces extremely unintuitive results, and explained how this arises from areas where the prediction function is non-additive ("attribution transfer"). This behavior may be unexpected because IG is founded in the theory of Shapley Values, an axiomatic framework for fair allocation. We explain this seeming contradiction by identifying a difference in the formulation of an axiom derived from the original theory.

In experiments we have shown that attribution transfer occurs even for state of the art methods like Expected Gradients in the presence of a highly skewed but plausible data distribution. Avoiding detrimental interactions by restricting IG to a small area of the input domain, as Additive Gaussian Noise does, does not prevent misleading results when the local prediction landscape does not reflect the wider input space.

We show these failures of IG may be avoidable, by developing a method with a new kind of baseline, Integrated Certainty Gradients, that requires a special training process to learn the data distribution. This method is the Aumann-Shapley alternative to SHAP (as IG is for BShap) and by not traversing the input space it avoids the failure cases we demonstrate in other methods.

ACKNOWLEDGMENTS

LEFT BLANK IN ANONYMOUS VERSION

ETHICS STATEMENT

Attribution is of key importance in many scenarios with ethical implications, for example providing reasons for life affecting decisions such as loan assessments, or determining whether decisions are made based on legitimate or protected personal characteristics. As such, the development of fair attribution methods is ethically important and this paper has been produced with appreciation of that. The work itself undertaken to produce this paper did not raise any ethical concerns.

REPRODUCABILITY STATEMENT

Model architectures and training hyperparameters are given in Appendix A.5. Anonymized source code is available at https://github.com/account48294/iclr1

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

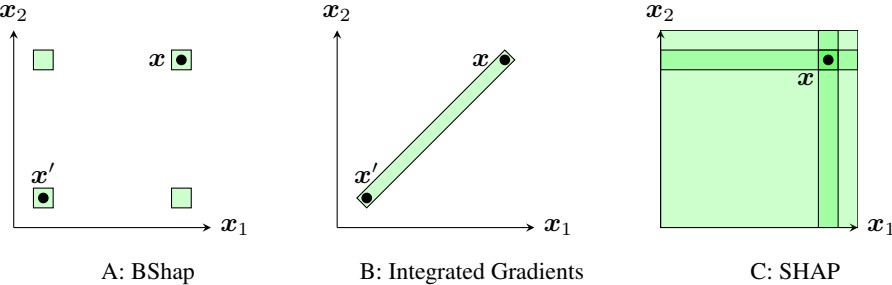

Figure 6: Input regions considered by attribution methods. Regions are shown with nonzero area for clarity, but in reality all regions except the whole-space region of SHAP are zero width points / lines of the input space. The input distribution is assumed uniform for SHAP. The locations of the target value $x$ and baseline $x'$ are arbitrary.

# A  APPENDIX

## A.1  ADDITIONAL THEORY AND CALCULATIONS

### A.1.1  SHAPLEY VALUES

Integrated Gradients is a method axiomatically founded in the theory of Shapley Values (Sundararajan et al., 2017). Shapley Values is a way to share responsibility between contributing factors when their combined outcome (expressed numerically) need not be equal to the sum of the outcomes each factor would give on their own. As such it is a way of sharing the benefit (or disadvantage) of cooperation. To do this it requires counterfactual knowledge of the outcomes that would be achieved by each subset of the factors alone. Formally: given a set $S$, and a value function $v : \mathcal{P}(S) \to \mathbb{R}$, with $\mathcal{P}(S)$ the power set of $S$, there is a Shapley value $\phi_{\text{sv}}(v, e)$ for each element $e \in S$. Its significance is that it is the unique method that, given this information, allocates responsibility according to the following axioms, intuitively associated with fairness:

**Efficiency** Also called Completeness. The Shapley values allocated to each of the factors sum to the value of the combined outcome. $v(S) = \sum\limits_{e \in S} \phi_{\text{sv}}(v, e)$

**Symmetry** Factors which have the same effect have the same Shapley value. Formally: if a pair of factors $e_1, e_2 \in S$ have $v(S' \cup e_1) = v(S' \cup e_2)$ for every $S' \subset S$ not containing $e_1$ or $e_2$, then $\phi_{\text{sv}}(v, e_1) = \phi_{\text{sv}}(v, e_2)$.

**Null player** Also called Dummy (Strong Dummy). If $v(S') = v(S' \cup \{e\})$ for all $S'$ then the Shapley value of $e$ is 0.

**Linearity** The Shapley Values method is a linear mapping of the value function: $\phi_{\text{sv}}(v_1, e) + \phi_{\text{sv}}(v_2, e) = \phi_{\text{sv}}(v_1 + v_2, e)$ and $\phi_{\text{sv}}(av, e) = a\phi_{\text{sv}}(v, e)$.

To calculate the Shapley values we define the *marginal contribution* of a factor $e$ to a set $S' \subset S$, $e \notin S'$ as $v(S' \cup \{e\}) - v(S')$. Then we consider every possible ordering of the factors in $S$. The Shapley value of $e$ is the mean of the marginal contributions it makes to the subset of elements that precede it in each ordering.

### A.1.2  BSHAP, SHAP, AND INTEGRATED GRADIENTS

Shapley Values assigns attribution to factors based on the differences in outcomes when they are absent. To represent missing components BShap uses the components of a baseline vector. The normal Shapley Values procedure is then followed, representing subsets of components by vectors. These vectors take values from the target vector $x$ for components in the subset, and from the baseline $x'$ for the other components. SHAP does not use a baseline. Instead it takes the expectation of the model output over the missing components, conditioned on the present components. In this paper we focus only on the model and so assume the input distribution uniform when considering SHAP. Therefore each subset in the Shapley Values calculation is represented by an integral across the

possible values of the absent components, with the other components fixed to their values in the target vector $\boldsymbol{x}$. IG does not follow the Shapley Values procedure but rather the method described in Section 1, based on the Aumann-Shapley method for sharing attribution. Figure 6 shows the regions of the input space used by each method.

### A.1.3 CALCULATIONS

The calculations for Table 1 follow. These follow the methods explained in the earlier parts of Section A.1. The model outputs are $F(\boldsymbol{x}_A) = 1$, $F(\boldsymbol{x}_B) = 0$, and $F(\boldsymbol{x}') = 0$.

**BShap**  With two features, there are two possible orderings for the Shapley calculations: $\boldsymbol{x}_1, \boldsymbol{x}_2$ and $\boldsymbol{x}_2, \boldsymbol{x}_1$. Starting from the baseline $\boldsymbol{x}'$ and swapping in components from the input $\boldsymbol{x}$ in these orders gives:

$$(\boldsymbol{x}_1', \boldsymbol{x}_2') \xrightarrow{\boldsymbol{x}_1} (\boldsymbol{x}_1, \boldsymbol{x}_2') \xrightarrow{\boldsymbol{x}_2} (\boldsymbol{x}_1, \boldsymbol{x}_2)$$
$$(\boldsymbol{x}_1', \boldsymbol{x}_2') \xrightarrow{\boldsymbol{x}_2} (\boldsymbol{x}_1', \boldsymbol{x}_2) \xrightarrow{\boldsymbol{x}_1} (\boldsymbol{x}_1, \boldsymbol{x}_2)$$

For B the value at all of these locations is zero so the attribution is zero for both components. Evaluating these for A:

$$F(0,0) = 0 \xrightarrow{\boldsymbol{x}_1} F(\boldsymbol{x}_1, 0) = 0 \xrightarrow{\boldsymbol{x}_2} F(\boldsymbol{x}_1, \boldsymbol{x}_2) = 1$$
$$F(0,0) = 0 \xrightarrow{\boldsymbol{x}_2} F(0, \boldsymbol{x}_2) = 1 \xrightarrow{\boldsymbol{x}_1} F(\boldsymbol{x}_1, \boldsymbol{x}_2) = 1$$

Which give attributions for $\boldsymbol{x}_1$: $\frac{1}{2}(1 - 1 + 0 - 0) = 0$ and $\boldsymbol{x}_2$: $\frac{1}{2}(1 - 0 + 1 - 0) = 1$.

**IG**  It is not necessary to perform an explicit calculation for IG. For A the path does not encounter 'vertical' gradients so the $\frac{\partial F(\boldsymbol{x})}{\partial \boldsymbol{x}_i}$ term in Equation 1 must be zero for $i$=2. Hence the second component has attribution 0. The attributions of all features must sum to the difference between the target input and baseline, giving the result $(1, 0)$. A similar argument can be developed for B by breaking the path into two parts and considering the traversal of the 'horizontal' and 'vertical' contours separately.

**SHAP**  There are again two possible orderings, this time of conditional expectations. Conditioned on both components $\mathbb{E}_{\mathbf{x}}(F(\mathbf{x})|\mathbf{x} = \boldsymbol{x})$ is just $F(\boldsymbol{x})$. Unconditioned $\mathbb{E}_{\mathbf{x}}(F(\mathbf{x})) \approx 0.434$ is the mean value over the whole input. Conditioned on only one component, the expectation integral is 1 if it is along one of the 'arms' of the cross, and 0.25 if it is across one of the arms. This gives attributions:

A:  $\boldsymbol{x}_1$: $\frac{1}{2}(1 - 1 + 0.25 - 0.474) \approx -0.11$.    $\boldsymbol{x}_2$: $\frac{1}{2}(1 - 0.25 + 1 - 0.434) \approx 0.66$.
B:  $\boldsymbol{x}_1$: $\frac{1}{2}(0 - 0.25 + 0.25 - 0.434) \approx -0.22$.    $\boldsymbol{x}_2$: $\frac{1}{2}(0 - 0.25 + 0.25 - 0.434) \approx -0.22$.

### A.2 PROOF THAT STRONG DUMMY IMPLIES CONSISTENCY

We define an attribution method $\varphi$ to be "Consistent" if given a model $F$, input $\boldsymbol{x}$ and feature $i$, an attribution $\varphi(F, \boldsymbol{x}, i) > 0$ implies there exists an $\boldsymbol{a}$ with $\boldsymbol{a}_i = \boldsymbol{x}_i$ such that $F(\boldsymbol{a}) > F(\boldsymbol{a} + \lambda\hat{\boldsymbol{x}}_i)$ for some $\lambda\hat{\boldsymbol{x}}_i$, a vector zero except for component $i$. That is to say, if a component is attributed positively, there must be at least one instance where changing that component of a vector to match the input vector results in an increase in value. This section demonstrates that adding Strong Dummy to the axioms Completeness, Symmetry, and Linearity guarantees Consistency.

BShap follows all of the four axioms. We will show it is the unique such method. BShap takes an input $\boldsymbol{x}$ and baseline $\boldsymbol{x}'$. It is evaluated using the Shapley Values procedure on all vectors $\boldsymbol{a}$ having $\boldsymbol{a}_i = \boldsymbol{x}_i'$ or $\boldsymbol{a}_i = \boldsymbol{x}_i$ for each $i$. Let $\mathcal{A}$ be a set of vectors so constructed. For a given $\mathcal{A}$ the axioms Completeness, Symmetry, Linearity, and Strong Dummy of an attribution method on the continuous space are equivalent to the Shapley Values axioms on the $\boldsymbol{a}$ space: Completeness requires the attributions sum to the total difference in output value between the baseline and target. This transfers directly to the Shapley Values context by subtracting the baseline value from every $F(\boldsymbol{a})$. Symmetry requires that the attribution method be invariant under permutations of the element/component labels; this is the same in both settings. In the continuous space the model functions $F$ are all the functions from input vectors to the possible attribution vectors. The functions relevant to the Shapley Values

perspective are all those from $\mathcal{A}$, a subset of the input space, to the same attribution vectors as before. Hence the functions on $\mathcal{A}$ are a linear subspace of the original function space, and so the linearity of an attribution method on the original space will apply to them also. Last, the definition of Strong Dummy was based on the Shapley Values Dummy axiom.

This demonstrates for any $x'$ and $x$, ie. everywhere, the four axioms are equivalent to the Shapley Values axioms. Given that Shapley Values is known to be the unique solution, BShap must be unique also. Shapley Values / BShap is consistent (if an element has positive value it must make at least one positive marginal contribution, a stronger guarantee) therefore Strong Dummy in combination with the other 3 axioms guarantees consistency. This discussion highlights the difference between Weak Dummy and the other axioms. Most of the axioms were strengthened when generalizing to the continuous setting, whereas Weak Dummy was weakened.

### A.3 CERTAINTY TRAINING

#### A.3.1 VALIDATING TRAINING

For effective training, methods to assess correct interpretation of certainty are required. We introduce two techniques for this.

**Information absence**    To ensure fair attributions, the model should interpret all baselines in the same way, as formalized by equation 2. This may be verified by varying $x$ with $c = 0$ and confirming predictions do not change.

Integrated Gradients gives zero attribution when components match the baseline (baseline blindness). However, Integrated Certainty Gradients relies on the model itself also ignoring zero *certainty* pixels, so that model and attribution behavior are aligned. This means that zero certainty pixels should not accrue attribution by any method, not just ICG. This makes $c = 0$ in a sense a universal baseline. This motivates a distinctive visual test. If the model is well trained, it should be possible to suppress the attribution of an accurate method to a negligible amount by masking out that area with zero certainty during the attribution process. This can be tested by using IG with baselines $x' = 0$ and $x' = 1$: the result should show an obvious suppression of attribution in the uncertain region in both cases. This test is visualized in Figure 7.

**Neutrality**    Zero *certainty* baselines should provide no input information to the model, so its output should be unbiased. It is therefore reasonable to expect the model prediction in the baseline space to match its expectation on the training set: $F(x, c = 0) \approx \mathbb{E}(F(X_{\text{train}})) \; \forall \; x$.

The expectation of each class in the training distribution is the proportion of samples in that class. So for example in one-hot representation, $\hat{y}_i$ should be approximately the proportion of the training distribution occupied by class $i$.

Using these tests we assessed a number of methods for certainty damage. The following methods improved training compared to using uniform distributions.

#### A.3.2 DAMAGE FUNCTIONS

The basic *damage function* described in section 4 can produce informative attributions but more effective *damage functions* have been identified.

**Variable damage extent**    With uniform random *certainty*, the trained model often widely violates the neutrality condition. Due to regression towards the mean, i.i.d. uniform random *certainty* results in an image with mean *certainty* $\frac{1}{n} \sum_i c_i$ typically close to 0.5. It is hypothesized that the the neutrality condition is not learned due to the lack of training inputs with mean *certainty* near zero. We define *damage extent* as the mean uncertainty of an input: $1 - \frac{1}{n} \sum_i c_i$. A *damage function* which results in *damage extents* distributed uniformly in [0, 1] will expose the model to inputs along more of the *certainty* interpolation space.

Sampling from the continuous Bernoulli distribution (Loaiza-Ganem & Cunningham, 2019) was chosen as a new *certainty* generating function because it produces samples in the range [0, 1] and has a parameter which allows the expectation to be varied in [0, 1] as well as other desirable properties.

Figure 7: Example of the Information Absence test. An area of the image is masked out to zero *certainty* during attribution (shown in grey). Integrated Gradients are taken from uniform zero and uniform one baselines. Even though these are not certainty-aware attribution methods, they show the same exclusion of attribution from zero certainty areas as ICG, suggesting that zero *certainty* is a "universal" baseline. Image E shows the attributions for the zero baseline and one baseline added together.

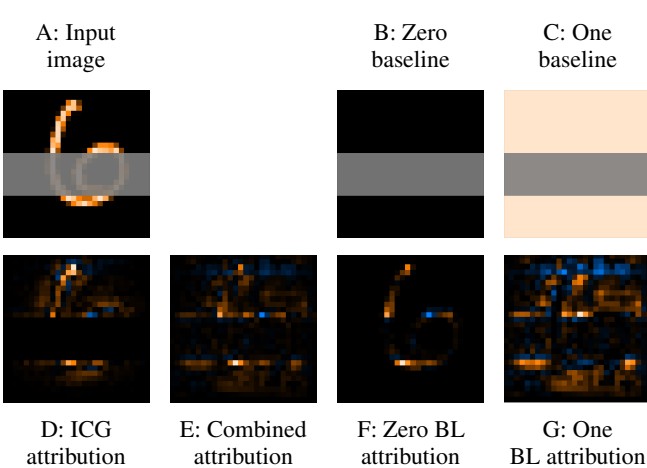

The parameter does not vary the expectation uniformly but instead concentrates it towards the center of the range. To counteract this we reparameterized the distribution. Let $e(\lambda)$ be the expectation of the distribution with parameter $\lambda$. Then $\lambda' := e^{-1}(\lambda)$, where $e^{-1}$ is calculated numerically from the analytical form of $e$. Using this new parameter gives a distribution parameterized by the mean. It was found that varying *damage extent* using this distribution does in fact improve neutrality.

**Adversarial damage**    Random noise is typically not strongly biased towards an input class. Uniform random *damage data* therefore results in an input which is a combination of data from the original class and unbiased data. As a result, the input is always positively biased towards the original class. Furthermore, if the model can detect and exclude random noise, it will be able to determine the true class from the remaining data. These considerations suggest the model might be able to ignore $c$ and still achieve good predictions.

Motivated by this, *adversarial damage* was tested: using a random alternative sample from the test dataset as the *damage data*. When the alternative and original input are from different classes, damaged features $D(x, c)_i$ provide evidence for a different class from the true one and with a large *damage extent* the majority of features can indicate the alternative class. It is not possible for the model to make an accurate prediction in such circumstances without considering $c$, forcing the model to use it. *Adversarial damage* was found to train certainty awareness faster and with more accuracy than random uniform damage. Another consideration is that it avoids exposing the model to out of distribution feature values, although the implications of this were not investigated.

**Binary uncertainty**    Continuous Bernoulli and uniform damage distributions have been described above. Discrete Bernoulli damage was also tested, producing $c_i$ values in $\{0, 1\}$. It was found that this improves learning rate compared to uniform random damage. The reason for this has not been determined, although it may be since $c_i$ can be only 0 or 1, the value components no longer require a probabilistic interpretation, which makes the scenario easier for the model to learn.

We found combining adversarial and uniform *damage data*, binary and uniform *uncertainty*, and using variable *damage extents* resulted in the most effective training in terms of speed and final evaluation. This training was used for all results which are discussed below.

### A.3.3    LEARNING RATE AND ACCURACY

*Certainty* awareness increases the classification task complexity. To assess the impact of this we trained MNIST models with and without input damage. The results are shown in Figure 8. The red and blue curves show training with and without damage respectively. Both are evaluated on undamaged inputs. They show that damage training has not resulted in reduced accuracy on MNIST for this model, although training takes twice as many batches. Variance in accuracy and loss is

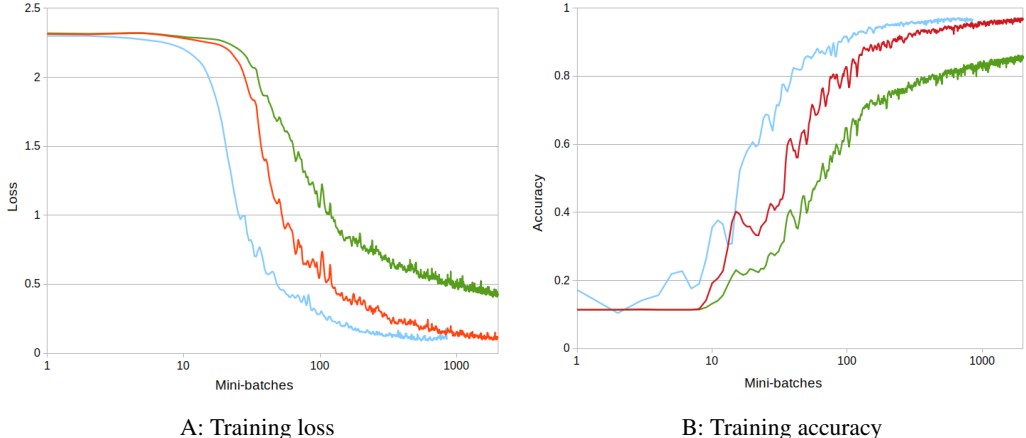

A: Training loss                    B: Training accuracy

Figure 8: Training progress with and without damaged inputs. Blue curves show learning progress without damage. Red curves show learning progress with damage evaluated on undamaged inputs. Green curves show learning progress with damage evaluated on damaged inputs. The evaluation images were drawn from the test dataset. Progress is measured in batches (of size 64) because multiple damaged inputs were created from each original input, leading to differently sized epochs. Evaluations on damaged inputs are significantly less accurate, which is to be expected given that when the *damage extent* is high they may contain little of the original image. Evaluations of the damage trained model on undamaged inputs reach a similar accuracy to the model trained without damage. Their curves show a roughly constant horizontal offset. On these logarithmic scales this indicates an approximately constant slowdown ratio: training with damage takes roughly twice as long.

also increased. Given that on average half the components of the original image are lost during damage, this seems a modest penalty. The green curve shows results for evaluation of the damage trained model on damaged inputs. This is included for completeness but not especially relevant to the comparison.

## A.4 ADDITIONAL RESULTS

### A.4.1 IMAGENETTE

To test ICG on more realistic scenarios, we trained a ResNet50 (He et al., 2015) using artificial uncertainty training on the Imagenette dataset (fast.ai, 2021). The only modifications made to the architecture were the addition of an extra channel to represent certainty, and an additional 1x1 (64 channel) convolution layer at the input, with ReLu activation and without batch normalization. The model was trained to 82.4% top-1 accuracy on the validation data.

Attributions using ICG were compared with those generated using EG, chosen because it is a contemporary baseline-free attribution method. It is established that single-path attribution methods such as ICG can produce clearer results when averaged over multiple slightly perturbed inputs (Smilkov et al., 2017). For comparison with EG, a multi-path method, we averaged 5 ICG attributions over inputs with added low-intensity noise to make each final result. We tested including noise augmentation when generating EG interpolations but it did not significantly affect the outcome.

Results can be seen in Figure 9. In most cases ICG attributions have a clear association with the object of interest. The attributions tend to be more tightly localized than those of EG. Backgrounds can form part of a model's classification decision, so stronger localization is not a full guarantee of greater attribution faithfulness. However, it is notable that in row 2 EG allocates significant attribution in the top black "border" region. It seems reasonable to assume this is one of the lowest information-density areas of the image, and not a representative region for attribution. We have not seen a case where ICG significantly attributes the border regions. Note that this is not baseline blindness: black/dark areas in images can be strongly attributed through ICG.

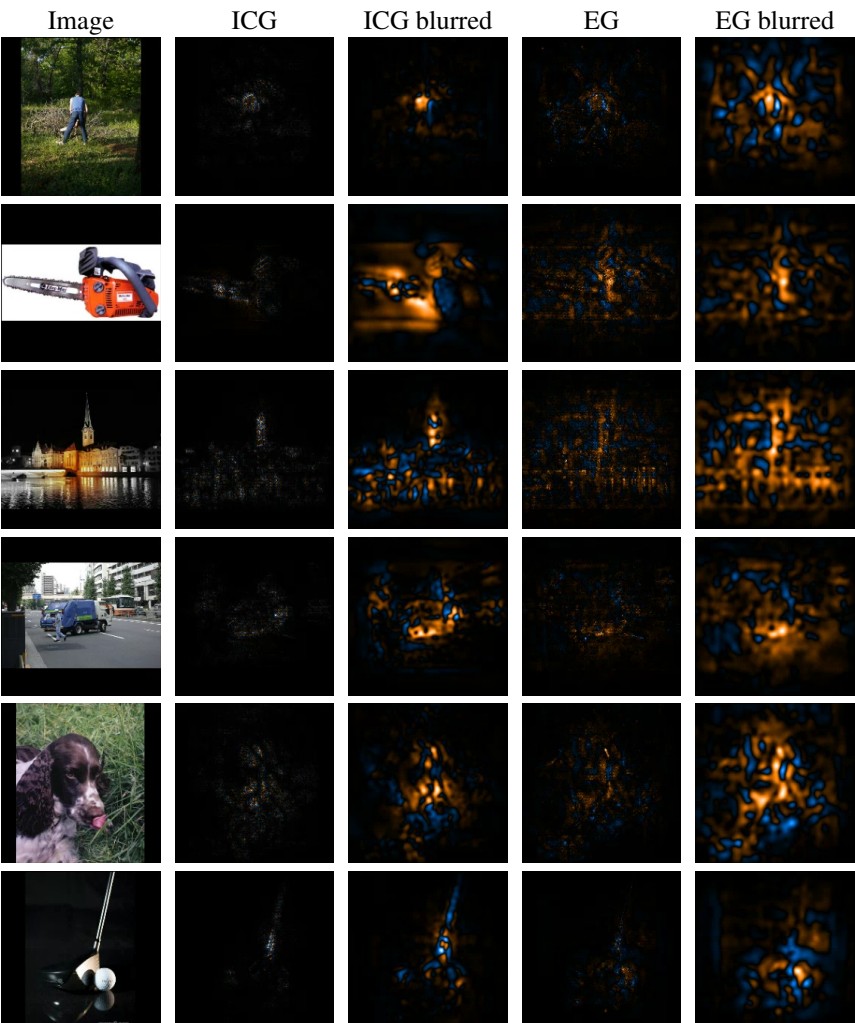

| Image | ICG | ICG blurred | EG | EG blurred |

Figure 9: Attribution results for Integrated Certainty Gradients and Expected Gradients applied to images from the Imagenette dataset. Images were randomly chosen discarding any incorrect classifications. Because methods in the Integrated Gradients family tend to produce "noisy"/"speckly" results, a Gaussian blur has been applied to each method to make interpretation easier. The results for ICG are notably more tightly localized in most cases. An exception is the chainsaw (row 2) where ICG strongly highlights background regions. In all other cases ICG clearly indicates the object of interest, except for the golf ball (row 6), where ICG focuses attribution primarily on the golf club.

We hypothesize the reason for the difference in behavior between EG and ICG. We have noticed output artifacts which relate to objects in individual EG interpolation baseline images. If EG were performed with enough interpolations to truly represent the input distribution it may be that the effects of interpolation baselines would exactly 'average out' in low attribution areas. It may be that this cannot be achieved in larger images due to the high dimensional input space. This would also explain the excellent performance of EG on the MNIST dataset (where the input distribution is rather densely represented). It would be reasonable to imagine that ICG might overcome this problem by approximating the input space as part of the deep learning model, given that deep learning methods are a current state-of-the art technology for representing high dimensional data.

Whether or our hypothesis is correct, the Imagenette attributions give evidence to ICG as an effective baseline-free attribution method in practical scenarios.

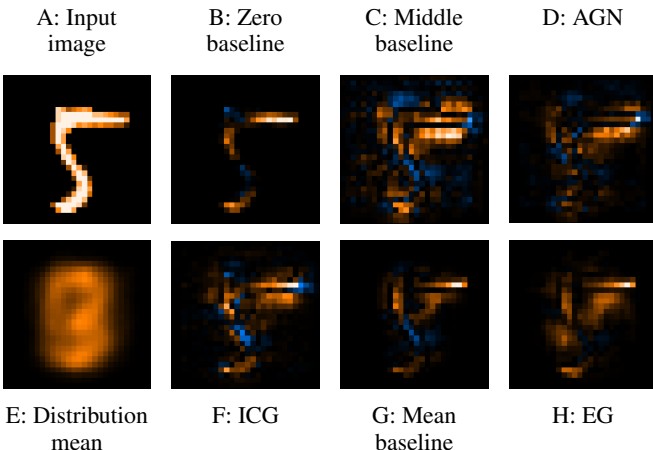

Figure 10: Attribution results for an example ("5") from the MNIST dataset (Lecun et al., 1998). A: Input image. E: Mean of the dataset. B-D: Integrated Gradients with baselines uniform 0, uniform 0.5, and Additive Gaussian Noise. F: Integrated Certainty Gradients. E: Integrated Gradients with dataset mean baseline. H: Expected Gradients. Positive values (orange) show supporting evidence. Negative values (blue) show opposing evidence.

### A.4.2 MNIST IN SPACE

Figure 10 compares attribution results for several baselines on a random example from the MNIST training test discussed in section A.3.3. The zero baseline attribution has strong baseline blindness: only the 'digit itself' is attributed, not the space around it. It shares several intuitively reasonable features with the other results. The crossbar at the top is positive, which is a characteristic feature of a "5". "7" is the only other numeral having one, and it terminates on the opposite side. The bottom curve is also indicated, a feature shared only with "3". The slope in the middle is more similar a 6 or an 8 (attributions for classes 6 and 8 attributions support this), and hence attributed negatively (in blue).

Of the remaining results there is a notable distinction between distribution-aware methods (mean baseline and EG) and unaware (middle baseline and AGN). For example in the unaware methods there is a negative attribution point where the crossbar terminates, indicating a preference for a longer crossbar, but not in the aware methods. This point is outside the region where the distribution varies, as can be seen from the distribution mean image, so it makes sense the distribution aware methods would not indicate it. ICG shares many characteristics with the distribution-unaware methods, but the most strongly attributed area, in the crossbar, matches to the aware methods. It is plausible it acts as an intermediate between the distribution aware and unaware methods: learning a distribution of the data during certainty training, but incorporating the model's generalization prior as it does so.

To explore this idea briefly we introduce a modified version of MNIST. We make the input images wider and introduce a translation-invariant model architecture. We can train a model using regular MNIST images in the left region, and the model is able to classify digits correctly anywhere in the extended image due to the architecture. When classifying digits in the right region they will be out-of-training-distribution. Results from this scenario are shown in Figure 11. The Expected Gradients result appears more sharply defined because of baseline blindness (the right side of all training images is black). A faint 'shadow' of the training distribution is visible in the EG result. This does not represent a model behavior since the model is translation invariant. EG is formulated as a fair baseline for a given distribution, but it does not follow the model's generalization onto OOD inputs. Because the distribution knowledge ICG uses is learned by the model, it is generalized by the model architecture and follows the model generalization onto new inputs. Like distribution unaware methods there is no 'shadow', because knowledge of the ungeneralized distribution is not retained.

### A.5 EXPERIMENTAL DETAILS

Model architectures are shown in Figures 14, 12, and 13.

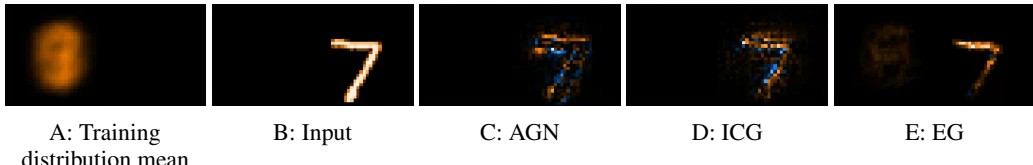

A: Training       B: Input       C: AGN       D: ICG       E: EG
distribution mean

Figure 11: MNIST in Space scenario. A: mean of the training inputs; B: target input for attribution; C: Additive Gaussian Noise attribution; D: Integrated Certainty Gradients attribution; E: Expected Gradients attribution. The main attribution of EG is significantly more localized due to baseline blindness. A 'ghost' of the training distribution is present in EG but not in AGN or ICG.

The *damage function* $D$ used during training is specified by a combination of *damage data* and *certainty* generator. For all models *damage data* was taken from either $U(0, 1)$ or *adversarially* from another training input. Within each input, $c_i$ values were taken i.i.d from either the Bernoulli or our reparameterization of the continuous Bernoulli distribution. Both have a parameter equal to their expectation, and for each image this was sampled from $U(0, 1)$. The four combinations of *damage data* selection method and *certainty* generation method were each used in equal proportion.

Xavier initialization was used for network weights and network biases were initialized to zero.

**Burnt snacks**    Training data was generated procedurally hence there is no epoch size or train-test split. Adam optimization was used with learning rate 0.001 and batch size of 64.

**Local minimum**    Training data was generated procedurally hence there is no epoch size or train-test split. The network did not converge for several training attempts, and was reinitialized several times until an initial state which did converge was found. The network was initially trained with Adam optimization and a learning rate of 0.001. After initial training, reduction in the learning was found to improve accuracy, and training was continued with reduced learning rates down to a minimum of $10^{-6}$. A batch size of 128 was used for initial training and a batch size of 256 for the reduced learning rate stage.

**MNIST**    The dataset was split into 60,000 training examples and 10,000 test examples across all classes. Adam optimization was used for training with learning rate 0.001 and batch size of 64.

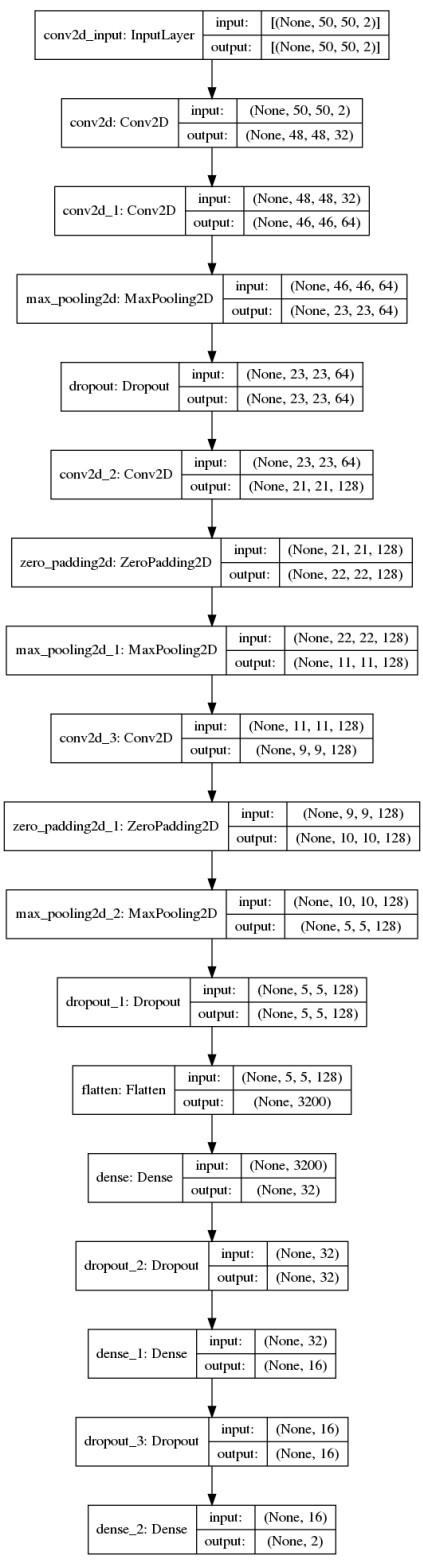

Figure 12: Model architecture used for Burnt Snacks scenario comparison.

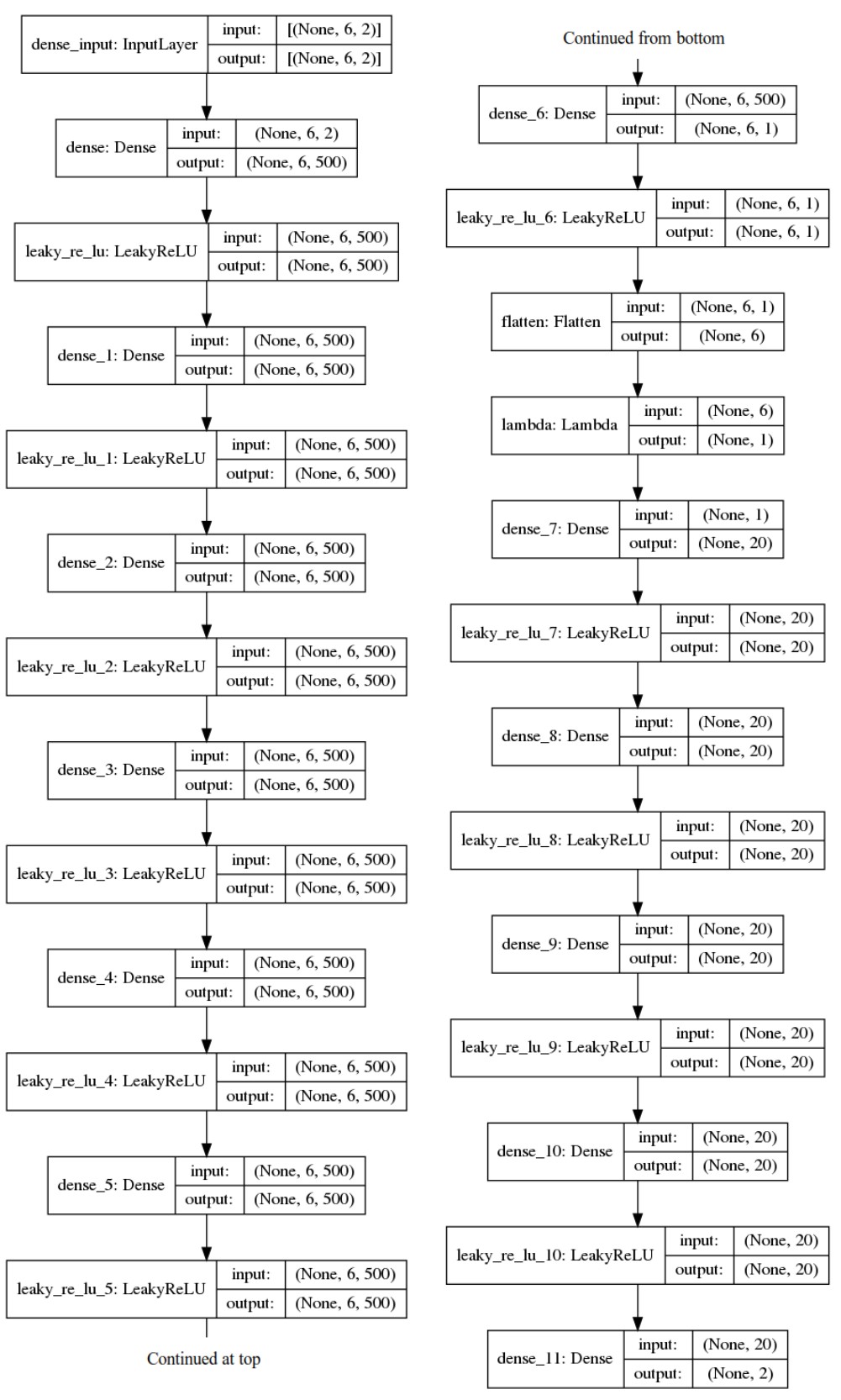

Figure 13: Model architecture used for Local Minimum scenario comparison. The layer 'lambda: Lambda' sums the input components.

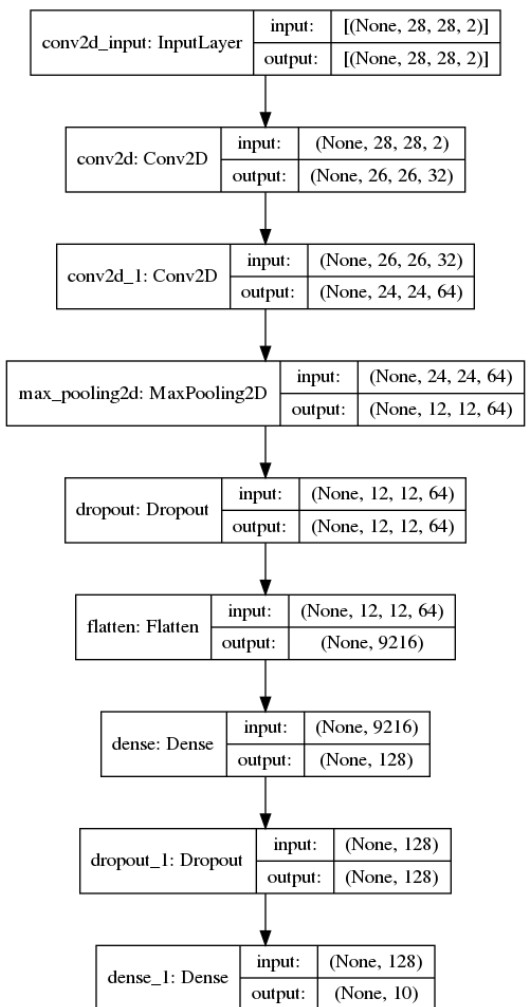

Figure 14: Model architecture used for MNIST testing.

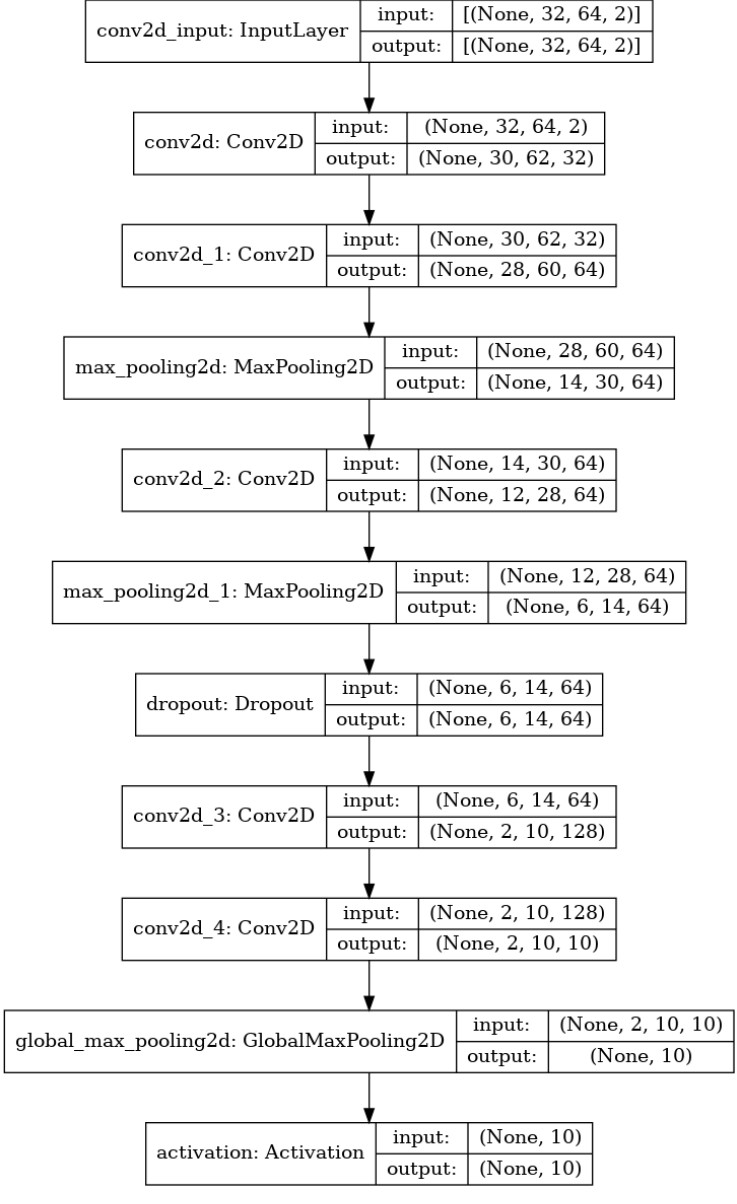

Figure 15: Model architecture used for MNIST in Space comparison.

