# OpenReview forum: "Exploring unfairness in Integrated Gradients based attribution methods"
_ICLR.cc/2022/Conference — ICLR 2022 Submitted_

### Official Review · Reviewer_W9En · 2021-10-30

**Correctness:** 3
**Technical Novelty And Significance:** 3
**Empirical Novelty And Significance:** 2
**Recommendation:** 5
**Confidence:** 3

**Main Review:**

For the discussion in Figure 2. What is the exact mapping, how the numbers in Table 1 were gotten. For path A, shouldn’t the attribution IG is less for x1 because the partial derivation is uniform along the horizontal axis?

For the Figure 2 example, I can’t understand very well what is ‘maximally misleading’ and what is ‘attribution functions’. Why IG is unfair? How to measure fairness.

For my understanding, SHAP and IG are both post-hoc attributions. But the proposed ICG needs to perturb the input by certainty noise (my words) to train the model. If this is correct, what is the performance of IG or SHAP on such a trained model? Any difference with the ICG?

Figure 3 also makes me confused. How to get the damaged input. What is the meaning of ‘sample’ here (bottom left). What do the green bounding box and blue bounding box mean? And blue and green rectangles as well.

I’m also wondering is there any visual difference for ICG and IG in terms of the visualization results on the real image like they show in their paper?

A typo, fourth line from the bottom of page 2, ‘any any’.

**Summary Of The Paper:**

The paper studies the unfairness of a family of integrated gradients based attribution methods, by the what they call ‘attribution transfer’. To solve the problem, it proposes a integrated certainty gradients attribution by adding ‘noisy’ to the input during the model training.

**Summary Of The Review:**

The theoretical analysis part (section 3) is somehow hard to understand for me. This lowers my score.

---

### Official Review · Reviewer_Ejjc · 2021-11-01

**Correctness:** 2
**Technical Novelty And Significance:** 3
**Empirical Novelty And Significance:** 2
**Recommendation:** 5
**Confidence:** 2

**Main Review:**

Strengths:
1. The proposed Integrated Certainty Gradients training and attribution framework is novel. It makes sense that the authors utilize the fact that the integrated gradient path will be potentially less fluctuated than passing through the input space when it is only calculated along the direction of certainty change. Therefore, the impact of the previous issues on the IG could be mitigated.

Detailed Comments:
1. In Section 3.1, the example of attribution transfer is somewhat ambiguous. This part could be more readable if the authors elaborate more on how BShap, SHAP, IG gave the exact attribution value, respectively.

2. In the first paragraph of Section 3.1, it is said that the model must indicate whether either of the two components falls within the range 0.25 - 0.5. However, from Figure 2 and its description, there is a differentiable region where the function value changes continuously (gradually from white to grey) at the edge of this range. I would suggest the authors modify this paragraph to be consistent with their figures.

3. The formal definition of attribution transfer seems unclear. In the third paragraph of Section 3.1, it is said that "We name this phenomenon attribution transfer" while "this" here is ambiguous. Based on the context, I infer that attribution transfer refers to "IG attribution results deviate from fair attribution due to the integration path traverses the fluctuating input space." However, I would like to see a more explicit definition given by the authors.

4. In Section 3.2, I think the logical coherence of the paper would be better if the authors could explain how the axiomatic difference is related to the attribution transfer and the fairness of IG.

5. In Section 4, it would be more convincing if the authors elaborate on the training process of ICG. I wonder whether there are changes compared with the ordinary training other than input data, such as loss function and backpropagation. Would ICG training affect the model performance? How and to what extend.

6. Two purpose-designed experiments without any general scenarios are insufficient to justify the use of Integrated Certainty Gradients. I notice that the authors mention their experiments on MINIST with little details in the appendix. I expect the attribution results of this experiment.

7. Since IG requires no modification of the original model while ICG trains the model with damaged data, it naturally makes me curious whether the utility of the ICG trained model decreases. I would suggest the authors show the test results of the ICG trained model and discuss whether there is a trade-off between generalization and attribution ability.

**Summary Of The Paper:**

In this paper, the authors analyze the fairness of Integrated Gradient-based attribution methods. They exploit SHAP and BShap, two approaches based on the theory of Shapley Values, as the reference of "fair" methods.

Specifically, they present an "attribution transfer" phenomenon in which the Integrated Gradients are affected by some sharply fluctuated area across the integration path, thereby deviating from the ''fair'' attribution methods. To avoid the attribution transfer issue, they further propose Integrated Certainty Gradients method, where the integration path does not pass through the original fluctuated input space.  Such an objective can be achieved by training the network with perturbed inputs and corresponding certainty maps. Finally, the gradients integral can be calculated by querying the trained network with fixed inputs and varying certainty. Various purpose-designed experiments are performed to demonstrate the advantages of ICG in avoiding attribution transfer.

**Summary Of The Review:**

Assessing fairness is important butt non-trivial. It is interesting to exploit SHAP and BShap, two approaches based on the theory of Shapley Values, as the reference of "fair" methods. Many concepts are not clear, and details are missing. Experiments are not comprehensive; please see the detailed comments.

---

### Official Review · Reviewer_FgJc · 2021-11-03

**Correctness:** 4
**Technical Novelty And Significance:** 2
**Empirical Novelty And Significance:** 1
**Recommendation:** 5
**Confidence:** 4

**Main Review:**

On the theory side:

This paper can been as a discussion on integrated gradients. The paper evaluates different baselines and moves to investigate the (un)fairness of attribution methods that the authors define as “whether the relative attributions of input components are reasonable”. After considering the (un)fairness of attribution methods in this respect, the authors propose their attribution method. The authors then apply their method to data. I did not find enough theory backing the usefulness of the proposed methodology in various aspects. Even though the paper has applied their methodology and has shown practice, the paper lack in theoretical support.

On the application side:

The paper applies their methodology to two scenarios in the main text and also provides the results of some experiments about accuracy on the MNIST dataset in the supplementary materials.

About the results in the main text, take Figure 4 as an example. The behavior of the proposed method is not clear. The method does not show attributions of the dark squares in columns B and C of the ICG row. Why is this the case? The authors have pointed out that repeating the algorithm leads to better results here, but the authors have not explained the reasons of this behavior clearly.

About the results of the supplementary materials, it is unclear how the results on the MNIST dataset can be evaluated and compared against other results in the main text. Also, the results in section A.1.3 are not promising.

Overall, the experiments of this paper are not convincing enough.

Looking at the theory and application together, the paper lacks novelty.

**Summary Of The Paper:**

This paper explores the characteristics of the method, Integrated Gradients, as an attribution method, that has been proposed to explain black box models. “Baselines” in analyzing integrated gradients are discussed and the shortcomings of integrated gradients are further evaluated. The paper then proposes Integrated Certainty Gradients and shows its application on data.

**Summary Of The Review:**

Even though this paper approaches an interesting problem, it needs improvements and clarifications.

---

### Author Response · Authors · 2021-11-17
**New revision submitted**

Dear reviewers,

We have submitted a new revision.

- The introduction to section 3, describing Shapley theory and attribution methods has been reworked for clarity.
- Section 3.1 has been updated
	- New figures have been added to make the basis for the "maximally misleading" claim easier to understand.
	- The description of the model has been improved to be more accurate.
	- "Attribution transfer" is defined.
- Section 3.2 has been extended to make the implications of the axiomatic assessment clearer.
- We have updated Figure 3 to explain the meaning of the colors.
- We have expanded upon the theory of Shapley values in new section A 1.1 of the supporting material.
- We have described the attribution methods in more detail in new section A 1.2 of the supporting material, including new figures.
- The calculations for the results in Table 1 are given in new section A 1.3 of the supporting material.
- A proof to support the axiomatic claims of section 3.2 is given in new section A 2 of the supporting material.

We are extremely grateful to each of the reviewers for the comments. Most of the improvements in this revision have been guided by this feedback.

We aim to make additional changes, and hope to address some of the remaining comments. If any reviewer has further feedback based on this version to inform our final assessment submssion, we would be extremely greatful for it. Particularly any concerns about the choice of content in the main text vs. supplemental material, or aspects of the work that remain unclear would be extremely welcomed.

Thank you for your help so far

---

### Author Response · Authors · 2021-11-23
**Third revision submitted**

Dear reviewers,

We have submitted a further revision.

In addition to the previous changes:

- We trained a ResNet50 architecture with certainty on the Imagenette dataset to demonstrate ICG in a realistic setting. Reasonable attribution results are generated. Compared to Expected Gradients we find the results more tightly localized to target objects.
  - 'Border' areas of images are respected, without the requirement for a baseline, nor baseline blindness in other parts of the image. EG has some attribution overflow into these areas.
  - We feel this gives evidence for ICG as an effective, fairly efficient, architecture independent (within the domain of artificial neural networks), baseline-free, blindness-free attribution method capable of producing state-of-the-art results.
- We provide an additional MNIST based scenario to clarify further some conceptual aspects of the ICG method.
- We identify the cause for the missing squares in the Burnt Snacks scenario and show it fits within the theoretical framework we have developed.

Regarding the significance of the theoretical aspect of the paper. Integrated gradients is a very important, widely used attribution method. It is well known as an axiomatically founded method, whose attributions have a strong association to the robustly fair Shapley Values axioms. To our knowledge no previous work has assessed whether Integrated Gradients offers similar guarantees. We find that it does not, and in fact results can deviate maximally from the fair behaviour expected of Shapley Values. We show this both analytically and experimentally. We further explore this behaviour theoretically and identify the specific axiomatic difference. We feel these results have wide relevance both to practitioners and researchers given they address such a fundamental aspect of the method.

We accept with the additions some aspects of the structure of the paper may not be perfect, particularly regarding the main text vs supporting material. We intend to improve these aspects. We will be very pleased to take on board any suggestions with respect to this (or any other) aspect for the final version, should the paper be accepted.

We sincerely thank the reviewers for their feedback, which we feel has helped us significantly improve the paper.

---

### Decision · Program_Chairs · 2022-01-20

**Decision:**

Reject

**Comment:**

This paper analyzes analyze the fairness of Integrated Gradient based attribution methods. The authors exploit SHAP and BShap, two approaches based on the theory of Shapley Values, as the reference of "fair" methods. Specifically, they present an "attribution transfer" phenomenon in which the Integrated Gradients are affected by some sharply fluctuated area across the integration path, thereby deviating from the ''fair'' attribution methods. To avoid the attribution transfer issue, they further propose Integrated Certainty Gradients (ICG) method, where the integration path does not pass through the original fluctuated input space. Experiments are performed to demonstrate the advantages of ICG in avoiding attribution transfer. While the basic premise of the work is interesting, many conceptual details remain unclear and experimental evaluation can also be improved (please see detailed reviewer comments below). Given this, we are unable to recommend acceptance at this time. We hope the authors find the reviews helpful.